# Food Import Dependency and National Food Security: A Price Transmission Analysis for the Wheat Sector

**DOI:** 10.3390/foods10081715

**Published:** 2021-07-23

**Authors:** Pengfei Luo, Tetsuji Tanaka

**Affiliations:** Department of Economics, Setsunan University, 17-8 Ikedanakamachi, Neyagawa, Osaka 572-8508, Japan; luo@setsunan.ac.jp

**Keywords:** import dependency, wheat, price volatility transmission, GARCH, DCC

## Abstract

Agricultural trade liberalization and protecting domestic markets encompass conflicting policy goals. Even though after the food crisis in 2008, national governments of food-deficit nations aimed at reducing food supply dependency on external markets, no research has assessed the impacts of food import reliance on price or price volatility transmissions to local markets. We constructed a dynamic conditional correlation (DCC)–generalized autoregressive conditional heteroscedasticity (GARCH) model to examine whether wheat import dependency could make a country vulnerable to overseas shocks by analyzing the inter-relationships between the international wheat price and retail wheat flour prices in 10 net importing countries over the sample period from January 2005 to December 2019. It was found that retail price volatility in each region was positively correlated with international price volatility for most of the period concerned. We also discovered that external dependency could significantly protect the domestic market from the global one, implying that lowering wheat dependency on foreign markets improves “stability” and “availability” of food security without sacrificing “utilization”, but it may aggravate “access”.

## 1. Introduction

Today, food insecurity in a region is often caused in remote areas through international trade. Over the last decades, this trend has been consolidated by multilateral or bilateral free trade agreements. Additionally, unconventional factors to jolt food markets such as hedge funds’ speculative money and biofuel production amplify international market volatility transmitted to vulnerable local markets in food-deficit countries. First-generation biofuel production limits arable land space, which could affect food markets. Furthermore, climate change induces extreme weather and destabilizes agricultural productivity and therefore market prices [1]. For example, a poor wheat harvest occurred due to extreme drought in Australia and Ukraine in 2007, which increased the global wheat price with a low level of global inventory, export restrictions, and financial speculative activities [2]. Thus, today, securing food supply from domestic or foreign markets is more difficult under such conditions.

Agricultural trade liberalization and protecting local markets have polarized policy goals. Since the 1980s, nations have signed several multilateral or bilateral free trade agreements to improve the efficiency of resource allocation, and countless citizens have received economic benefits from trade. In the wake of the 2008 global food crisis that worsened global food insecurity, extending the hunger population by 44 million [3], many national governmental bodies, such as those of Senegal, India, the Philippines, Qatar, and Bolivia, expressed interest in food self-sufficiency policy [4]. In other words, various countries are attempting to establish robust food supply systems to prepare for the uncertainty that could happen in agricultural international markets. Food autarky is broadly regarded as an effective strategy for national food security by policymakers. This policy measure could reduce the degree of international price transmissions, which may interest the aforementioned governments.

The existing literature regarding price transmission of food commodities is abundant, with over 500 recently published papers resulting from an AgEcon database search with the term “price transmission” [5]. Many papers analyze price pass-throughs within developing countries (e.g., [6,7,8,9,10,11,12,13]). Fewer papers explore price correlations from global to local markets (e.g., [14,15,16,17,18]), many of which apply error correction models to establish the international market connections. The only paper that investigates international price volatility transmissions in agricultural sectors was by Ceballos et al., in which a bivariate threshold Baba, Engle, Kraft and Kroner (BEKK)- generalized autoregressive conditional heteroscedasticity (GARCH) model is used [19]. However, most of the price transmission analyses mentioned above fail to explore the potential determinants behind international price pass-throughs. Guo and Tanaka examined the effectiveness of self-sufficiency in wheat for wheat importers and exporters, respectively [8,20]. They discovered that a high autarky rate has the potential to weaken price volatility transmissions between global and local markets.

This paper focuses on the identification of potential factors behind international price volatility transmissions of wheat, using GARCH- dynamic conditional correlation (DCC) models over the period from January 2005 to December 2019. We first estimated the time-variant relationship between world and regional prices, and then we regressed the estimated DCC or asymmetric dynamic conditional correlation (ADCC) by potential transmission factors such as wheat import dependency, gross domestic product (GDP), consumer price index (CPI), exchange rates, and the consumption of substitutive goods such as rice, corn, and barley.

The contribution of this research to the literature is to discover the potential determinants of international price volatility pass-throughs concentrating on net wheat import or dependency on foreign suppliers. Even though there exists a large body of literature on agricultural price transmissions as stated above, only a few articles attempt to find the potential drivers behind price or price volatility transmissions [8,20,21]. The past research concentrates on self-sufficiency policy in agricultural commodities, but no article discusses the role of food import dependency in the connectedness between global and local markets. The outcomes from our experiments are useful for policymakers to prevent unexpected price fluctuations of local food markets in terms of national food security.

This paper is organized as follows. The “Methodology” section describes the empirical models and data used in the present paper. The next section presents the results, interpretations of the outputs, and policy implications. Finally, the “Conclusion” section summarizes the article and mentions future research directions.

## 2. Materials and Methods

We used a two-step econometric methodology to analyze the price volatility transmission between global and local wheat prices. First, we employed a dynamic conditional correlation–generalized autoregressive conditional heteroscedasticity (DCC-GARCH) model to estimate the volatility spillovers between global and local wheat prices. Second, a panel analysis was applied to investigate the common factors affecting the volatility spillovers between global and local wheat prices.

### 2.1. Estimating Volatility Transmission Using DCC-GARCH Models

Since Bollerslev introduced the generalized autoregressive conditional heteroscedasticity (GARCH) process to forecast univariate financial asset volatilities, the theory of multivariate GARCH (MGARCH) models has developed to reveal the co-volatilities of multiple asset prices [22]. Among these, the DCC-GARCH model developed by Engle represents a computational advancement when estimating large covariance matrices to describe the time-varying correlation between asset prices [23]. This paper, therefore, employs a DCC-GARCH model to analyze the volatility transmission between global and local wheat prices.

In a DCC-GARCH model, the mean equation is
(1)$$p_{t}=c+\sum_{i=0}^{k}a_{i}p_{t-i}+u_{t}$$
where *p_t_* is a 2 × 1 vector including the logarithmic returns of global and local wheat prices, *p*_1,*t*_ and *p*_2,*t*_, respectively; *c* is a 2 × 1 constant vector where *c* = (*c*_1_,*c*_2_)′; *a_i_* is the coefficient which captures the autoregressive effect. The lag length *k* could be determined by an information criterion; *u_t_* is a 2 × 1 forecast errors vector where *u_t_* = (*u*_1,*t*_,*u*_2,*t*_)′. 

The variance equation follows a DCC-GARCH process as
(2)$$\left\{\begin{array}{c} \begin{array}{c} u_{t}=\sqrt{H_{t}\varepsilon_{t}}\\ H_{t}=D_{t}R_{t}D_{t} \end{array}\\ D_{t}=diag\left(\sqrt{h_{11,t}},\sqrt{h_{22,t}}\right)\\ R_{t}={\left(diag\left(Q_{t}\right)\right)}^{-\frac{1}{2}}Q_{t}{\left(diag\left(Q_{t}\right)\right)}^{-\frac{1}{2}}\\ Q_{t}=\left(1-\lambda_{1}-\lambda_{2}\right)\bar{Q}+\lambda_{1}{\tilde{u}}_{i,t-1}{\tilde{u}}_{j,t-1}+\lambda_{2}Q_{t-1} \end{array}\right.$$
where *H_t_* is a 2 × 2 conditional variance–covariance matrix; *ε_t_* is a 2 × 1 vector of normal, independent, and identically distributed innovations; *R_t_* is a 2 × 2 symmetric dynamic correlations matrix; *D_t_* is a 2 × 2 diagonal matrix of conditional standard deviations for forecast residuals; Q*_t_* is a 2 × 2 time-varying covariance matrix of standardized residuals *ũ_i_*_,*t*_ = *u_i_*_,*i*_/*h_i_*_,*t*_; and $\bar{Q}$ is the unconditional correlation as *ũ_i_*_,*t*−1_*ũ_j_*_,*t*−1_. Constraints that both *λ*_1_ and *λ*_2_ are non-zero and *λ*_1_ + *λ*_2_ < 1 to keep the volatility convergence are set. If *λ*_1_ + *λ*_2_ = 1, it becomes a constant conditional correlation GARCH model (CCC-GARCH), which has no conditional variance and covariance between prices. The computational advancement of the DCC-GARCH model is that conditional variances *h*_11,*t*_ and *h*_22,*t*_ in *H_t_* are estimated by a univariate GARCH process as
(3)$$h_{i,t}=b_{i,0}+b_{i,1}u_{i,t-1}^{2}+b_{i,2}h_{i,t-1}$$
where *b_i_*_,0_, *b_i_*_,1_, and *b_i_*_,2_ are coefficients.

Furthermore, ref. [24] introduced the asymmetric DCC-GARCH (ADCC-GARCH) model, which modifies the correlation evolution equations with an asymmetry as
(4)$$Q_{t}=\left(1-\lambda_{1}-\lambda_{2}\right)\bar{Q}-\delta \bar{N}+\lambda_{1}{\tilde{u}}_{i,t-1}{\tilde{u}}_{j,t-1}+\lambda_{2}Q_{t-1}+\delta (\eta_{t-1}{\eta^{\prime }}_{t-1})$$
where $\bar{N}$ represents the unconditional matrices of $\eta_{t}=I\left[{\tilde{u}}_{i,t}<0\right]⊗{\tilde{u}}_{i,t}$ where ‘⊗’ is the Hadamard product; *I*[.] is an indicator function equal to 1 if *ũ_i_*_,*t*_ < 0 and 0 otherwise; *λ*_1_ and *λ*_2_ have the same definition and constraints as in the DCC-GARCH model.

The parameters in DCC-GARCH and ADCC-GARCH models are estimated using the quasi-maximum likelihood method (QMLE). Under the Gaussian assumption, the log-likelihood function is

(5)$$\mathrm{LL}=-\frac{1}{2}\sum_{1}^{2}\left[2\log\left(2\pi \right)+\log{\left|D_{t}\right|}^{2}+u_{t}^{\prime }D_{t}^{-1}D_{t}^{-1}u_{t}\right]+\left(\log\left|R_{t}\right|+{{\tilde{u}}^{\prime }}_{t}R_{t}^{-1}{\tilde{u}}_{t}-{{\tilde{u}}^{\prime }}_{t}{\tilde{u}}_{t}\right)$$

### 2.2. Estimating Common Factors for DCC Using Panel Analysis

In the next step, we analyzed which common factors affect the volatility transmission between global and local prices by using a panel analysis as
(6)$$DCC_{i,t}=\mu_{0}+\mu_{1}Dependence_{i,t}+\mu_{2}GDP_{i,t}+\mu_{3}CPI_{i,t}+\mu_{4}\Delta er_{i,t}+\sum_{j=1}^{3}\gamma_{j}\Delta SC_{j,t}+d_{year}+d_{country}+\varepsilon_{i,t}$$
where *DCC* is the annualized dynamic conditional correlations between global and local wheat prices calculated by DCC-GARCH models; *Dependence* is the degree of dependence on wheat import calculated by $Dependence_{i,t}=\frac{Net\;Import_{i,t}}{wheat\;consumption_{i,t}}\times 100$; *GDP* is the annual GDP per capita growth rate; *CPI* is the annual inflation rate; ∆*er* is the logarithm exchange rate returns of the local currency against the US dollar; ∆*SC* is the logarithmic change rates of consumption in substitute goods, including barley, maize, and rice; *d_year_* and *d_country_* are year and country dummies; *ε_i_*_,*t*_ is the heteroskedastic error term; *μ* and *γ* are coefficients to be estimated.

Table 1 shows the expected signs of coefficients estimated in Equation (6).

We assume the following: (1) A higher degree of dependence on wheat imports and the local currency depreciation against the US dollar will weaken the volatility transmission between global and local wheat prices, resulting in a higher *DCC*. (2) Inflation in local prices and increases in consumption of substitute goods will strengthen volatility transmission between global and local wheat prices. More importantly, as wheat is a necessary good, the price elasticity to incomes is assumed positive.

### 2.3. Data

This analysis concentrates on assessing the impacts of import dependency in wheat on price volatility transmissions between global and local markets. Accordingly, we chose only net importers of wheat with long time series data to extend the sample size. Since wheat is not directly consumed by households, wheat flour prices were collected for domestic prices. We gathered data from the Global Information and Early Warning System (GIEWS), which offers monthly agricultural commodity prices by country. Retail flour prices from January 2005 to December 2019 for 10 wheat-importing countries (Armenia, Brazil, Cameroon, Georgia, Kyrgyzstan, Peru, South Africa, Japan Mexico, and South Korea) were obtained. Commodity price data for many nations in the GIEWS are available, but we found only 10 regions that meet our requirements, such as the availability of data on retail prices of wheat flour for a recent period and being non-self-sufficient in wheat. Following the literature (e.g., [20]), both the global wheat price and local wheat prices were invoiced in the US dollar. The global wheat price is the export price of wheat (the US No. 2 hard red winter) from the US Gulf Coast, which is also quoted in the GIEWS. Table 2 shows the variables employed in panel analysis.

## 3. Results

### 3.1. Dynamic Conditional Correlations between Global and Local Wheat Prices

Figure 1 shows the movement in the logarithmic returns of global and local wheat prices. Both global and local wheat prices volatilize mainly within a range of ±20% during the entire sample period and experienced large price volatility after the 2008 Global Financial Crisis.

Table 3 reports the descriptive statistics and the augmented Dicky–Fuller (ADF) test statistics of wheat price returns. Both the global price and local prices in most importer countries averagely increased from January 2008 to December 2018, except South Korea and Peru. The augmented Dicky–Fuller (ADF) test on global and all local price returns rejects the null hypothesis that there is a unit root at a 1% significant level, showing stationarity at the level.

In addition, we applied the Granger causality test between global and local price returns by country to find the causality with one- or two-period (month) lag. The results in Table 4 shows that, for most wheat importer countries, there is significant Granger causality from the global price return to local price returns, expressing the large dominant power of the global price on local prices. For Armenia, Brazil, and South Korea, there are significant Granger causalities from local price returns to global returns, showing the local prices of importer countries could even affect the global price. Conversely, for Japan and South Africa, there are no significant Granger causalities.

Regarding the stationarity and Granger causality of sample data, we estimated four types of DCC-GARCH models, as mentioned in the methodology: the DCC-GARCH model, the DCC-GARCH model with autoregressive effect in mean equation (AR-DCC-GARCH), the ADCC-GARCH model, and the ADCC-GARCH model with autoregressive effect in the mean equation (AR-ADCC-GARCH). Table 5 reports the DCC-GARCH model selection by Bayesian information criterion (BIC). It is notable that, for some model settings, dynamic conditional correlation could not be estimated by the maximum likelihood method, possibly due to the small sample size. After excluding the inestimable model settings (represented by a hyphen in Table 5), the best model selections for countries were chosen by BIC: DCC-GARCH models for Cameroon, Japan, and Kyrgyzstan; an AR-DCC-GARCH model for South Korea; ADCC-GARCH models for Brazil, Georgia, and South Africa; AR-ADCC-GARCH models for Armenia, Mexico, and Peru.

Table 6 presents the estimation results of the parameter metrics for the DCC models with specifications selected by BIC, which are shown in Table 5. First, every model specification fits the condition of *λ*_1_ + *λ*_2_ ≤ 1, indicating that the dynamic conditional correlation for all pairs of global and domestic wheat prices is mean reverting (convergent). Second, the coefficient *λ*_2_ is observed to be positive and significant for most model specifications, showing the autoregressive effect that the lagged dynamic conditional correlation significantly affects the current dynamic conditional correlations. Third, the asymmetric coefficients, *δ*, are found to be positive, indicating the asymmetric response in the correlation between global and domestic wheat prices, although it is not statistically significant at a 10% significant level for all selected model specifications possibly due to the limitation on data size.

Figure 2 represents the estimated dynamic conditional correlations between global and domestic prices, following the model specifications selected by BIC.

Overall, the movements in dynamic conditional correlations show strong time variability in the sample period and considerable country dependence. For almost the entire sample period, dynamic conditional correlations are positive for sample countries, showing that the high volatility of the global wheat price could lead to an increase in the volatility of local wheat prices. Further, we also found that the dynamic conditional correlations could become negative in a very short time after shocks. For example, the DCC for Japan became negative after the 2008 Global Financial Crisis, showing the different movements between global and local prices. The same scenario also happened in Kyrgyzstan and South Korea after the shock in 2017. Therefore, shocks could affect the conditional correlation between global and local prices, showing the different responses of global and local wheat markets. Finally, we found a significant time trend of the dynamic conditional correlation in some sample countries such as Armenia, Cameroon, and Japan. For example, the DCC in Cameroon experienced a sharp decrease after the 2008 Global Financial Crisis and then stabilized around 0.3 from 2012 to 2018.

Further, Table 7 indicates the descriptive statistics of monthly DCC between global and local wheat prices. We found that, during the sample period, all country DCCs have positive means, displaying the increasing volatility spillovers between global and local wheat prices in import countries. The ranges of DCCs largely varied depending on the country. For example, the Armenian DCC range was 0.068, while the Mexican DCC range was 0.759, showing the different movements of DCC.

Figure 2 signifies the DCC pairs between the global wheat price and retail wheat flour prices. It is observed that the DCCs for all regions over the sample period never take negative values, except those for Japan in the food crisis (2007–2008) period and Kyrgyzstan and South Korea around 2017. This suggests that greater price volatility in the global market tends to cause larger price volatility in local markets. It is interesting that the dynamic correlations dramatically fluctuate around the 2008 food price surges in most of the nations (Armenia, Brazil, Cameroon, Mexico, South Korea, Peru, and South Africa), while the connectedness largely varies over time and across regions. The mean of the correlations is positive in all regions. Cameroon has the highest average estimate of the DCC (0.306), while Armenia shows the lowest value (0.072).

### 3.2. Panel Data Analysis

Table 8 exhibits the results of the panel data analysis. It is indicated that the coefficients of dependency on foreign markets are positive and significant for all models following the hypothesis. The experimental outcomes confirm that the net import of wheat is a potential determinant for international price transmission and that larger foreign supply reliance induces greater spillovers to local markets. We also found that CPI is positive and significant for DCC in all models. This implies that international market connectivity is strengthened by high inflation. This is because, when the relative domestic wheat or wheat flour price to the counterpart in overseas markets becomes high, consumers prefer foreign products, leading to an increase in wheat imports. As such, the importing retail prices are more sensitively influenced by the global price. Moreover, the outcomes tell us that substitutive goods such as barley, maize, and rice do not significantly function as a shock absorber for domestic wheat flour markets, which differ from the hypothesis.

### 3.3. Policy Implications

Our results indicate that reducing the foreign dependency on wheat could significantly abate the conveyance of international market shocks to indigenous markets. This partially bolsters the efficacy of the food self-sufficiency policy that gained the attention of food-deficit nations especially after the 2008 food price spikes. For instance, since the Second World War, the food self-sufficiency of Japan on a calorie basis has almost halved to approximately 40%, and the central government has long attempted to push up its food autarky rate by protecting domestic markets from global markets. Because the wheat self-sufficiency in Japan is achieved with around 10% imported from major exporters such as the US, Australia, and Canada, reducing reliance on foreign suppliers could stabilize local prices. After the global commodity boom in 2008, many developing countries suffered the price spikes of bread, a primary diet commodity, and people sparked off food riots [25]. Egypt, the largest importer of wheat, experienced food riots in 2011 and 2017 [26,27], which may have been prevented with the policy measure.

Due to the discovery of cows with bovine spongiform enteropathy in the United States around 2004, Japan’s government restricted beef imports from the country, which caused substitutive behavior with many restaurants replacing beef meals with pork ones immediately after the event. In this context, our analysis uncovered that substitutive grain consumption behavior was not useful for alleviating shocks from external markets. However, the result explains only the short-term impact, and long-term effects may be different. Namely, cereal substitution could occur in the long run, mitigating price volatility transmission from world markets. However, the diversification of food consumption patterns must be beneficial in terms of financial risk management for households. Consumers who have eating habits of just wheat-based products are economically susceptible to wheat price fluctuations, compared with those who consume a variety of grain-based food products. Japan is a good example of diversifying food goods. Before the end of the Second World War, the citizens of Japan depended only on traditional Japanese meals. This was largely changed with the introduction of Western-style foods to school lunches by the General Headquarters of the Supreme Commander of Allied Powers that reigned over the nation. Consequently, the population in the country consumes a wide range of international meals today. Although it would take a long time to actualize food consumption diversity, the policy implementation cost to maintain the eating habits of nationals does not seem to be large once it is realized.

Food security comprises four aspects: “availability”, “access”, “stability”, and “utilization”. From these viewpoints, our results suggest that, assuming that domestic wheat production is boosted by, for instance, a higher import tariff or additional farming subsidy, the reduction of food supply dependency improves “availability”. The resulting lower external supply dependency could improve “stability”. From the perspective of farmers, the reduction of price volatility under the autarky system improves their price forecast, which increases farmers’ expected utility levels.

## 4. Conclusions

This paper examines the potential determinants of international price volatility pass-throughs for the wheat sector, applying DCC-GARCH and panel data models. Our primary results are as follows: (1) the global wheat price and the retail price of wheat flour are positively correlated over time; (2) lowering reliance on foreign suppliers weakens the connectedness between global and local prices; (3) substitutive grain commodities such as barley, maize, and rice do not absorb jolts from overseas markets.

This research focused on the identification of cross-boundary price volatility transmission and its underlying factors. In policy decision-making processes, the implementation cost needs to be estimated to compare the benefits from the policy. As stated, the central government of Japan could not enhance food autarky rates even when allocating a considerable amount of subsidy. Even if self-sufficiency were hoisted, the policy should not be carried out if the comprehensive beneficial effects are smaller than the financial burden. Therefore, the favorable impacts from stabilizing domestic markets also need to be assessed. These subjects are beyond the scope of this paper and left for future research.

## Figures and Tables

**Figure 1 foods-10-01715-f001:**
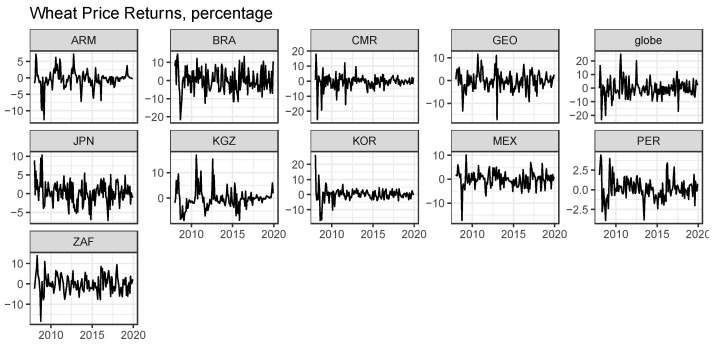
The movements in global and local wheat prices returns. ARM, BRA, CMR, GEO, globe, JPN, KGZ, KOR, MEX, PER, and ZAF stand for Armenia, Brazil, Cameroon, Georgia, global, Japan, Kirgizstan, South Korea, Mexico, Peru, and South Africa, respectively.

**Figure 2 foods-10-01715-f002:**
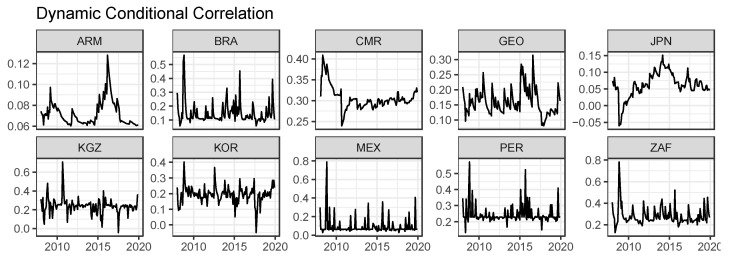
The movements in volatility transmissions. The panels show the movements in dynamic conditional correlations between global and local wheat prices calculated by the DCC-GARCH model.

**Table 1 foods-10-01715-t001:** Assumptions on panel analysis results.

Dependent Variables	Dependence	Macroeconomic Factors			Changes in Consumption of Substitute Goods		
		*GDP*	*CPI*	∆*er*	∆*SC_barley_*	∆*SC_maze_*	∆*SC_rice_*
Expected signs	+	+	−	+	−	−	−

Note: This table shows the expected signs of the coefficients in panel analysis on *DCC* by Equation (6).

**Table 2 foods-10-01715-t002:** Panel analysis variable descriptions.

Variable	Descriptions	Source
*DCC*	Annualized dynamic conditional correlation between logarithmic returns of international and local wheat prices (unit: percentage)	Authors’ calculation
*Dependence*	Degree of dependence of local wheat consumption on import, calculated by $\frac{net\;wheat\;import}{local\;wheat\;consumption}\times 100$ (unit: percentage)	FAOSTAT; authors’ calculation
*GDP*	Annualized growth rate of gross domestic product per capita (unit: percentage)	FSI
*CPI*	Customer price index (unit: percentage)	FSI
∆*er*	Logarithmic return of exchange rate of local currency against the US dollar (unit: percentage)	FSI; authors’ calculation
∆*SC*	Logarithmic return of consumption of substitute goods, including barley, maize, and rice (unit: percentage)	FAOSTAT; authors’ calculation

**Table 3 foods-10-01715-t003:** Descriptive statistics and augmented Dickey–Fuller test statistics of wheat price returns.

Variable	Mean	Median	Minimum	Maximum	Standard Deviation	Skewness	Kurtosis	ADF
ARM	−0.33	−0.16	−12.71	7.24	2.70	−0.90	4.61	−6.30 ***
BRA	0.05	−0.65	−21.59	14.53	6.21	−0.13	0.19	−7.98 ***
CMR	−0.35	−0.08	−25.56	17.91	4.66	−1.30	8.50	−11.40 ***
GEO	−0.13	−0.18	−17.18	11.60	3.94	−0.47	2.69	−6.95 ***
Globe	−0.36	−0.88	−23.20	24.97	6.55	0.30	2.52	−9.52 ***
JPN	0.25	0.32	−7.15	10.36	2.91	0.32	1.11	−6.10 ***
KGZ	−0.25	−0.37	−8.69	16.92	3.95	1.20	3.82	−5.14 ***
KOR	0.01	0.20	−17.08	26.22	4.19	0.74	13.09	−8.57 ***
MEX	−0.06	0.10	−17.20	10.11	3.21	−0.82	4.91	−7.99 ***
PER	0.11	−0.06	−3.90	4.43	1.36	0.35	1.28	−7.26 ***
ZAF	−0.07	−0.18	−18.29	13.73	4.30	−0.14	1.68	−7.32 ***

Note: *** *p* < 0.01.

**Table 4 foods-10-01715-t004:** Granger causality test.

**Lag**	**Armenia**		**Brazil**		**Cameroon**		**Georgia**		**Kyrgyzstan**		**Peru**	
	**1**	**2**	**1**	**2**	**1**	**2**	**1**	**2**	**1**	**2**	**1**	**2**
global→local	22.38 ***	13.93 ***	4.63 **	3.38 **	0.87	1.18	23.57 ***	14.49 ***	25.81 ***	12.87 ***	9.22 ***	5.18 ***
local→global	0.12	2.35 *	2.61	2.67 *	2.5	2.43 *	0.24	0.67	0.14	0.72	0.06	1.13
**Lag**	**South Africa**			**Japan**				**Mexico**		**South Korea**		
	**1**		**2**	**1**		**2**		**1**	**2**	**1**	**2**	
global→local	0.91		1.13	0.08		0.19		3.16 *	5.96 ***	6.04 **	3.77 **	
local→global	1.48		2.2	1.15		0.63		0.38	0.33	4.94 **	3.46 **	

Note: *** *p* < 0.01, ** *p* < 0.05, * *p* < 0.1.

**Table 5 foods-10-01715-t005:** The model selection of four specifications for DCC GARCH models.

GARCH Model	ARM	BRA	CMR	GEO	JPN	KGZ	KOR	MEX	PER	ZAF
AR–ADCC	11.352 *	13.404	12.335	12.519	11.904	-	12.154	12.026 *	10.041 *	-
ADCC	11.543	13.344 *	12.339	12.476 *	11.879	12.239	12.089	12.041	10.044	12.525 *
AR–DCC	-	-	12.333	12.487	11.87	-	12.015 *	-	-	-
DCC	-	-	12.313 *	-	11.845 *	12.204 *	11.967	-	-	-

Note: * denotes the best model selection by BIC criterion, while - denotes no dynamic conditional correlation could be calculated.

**Table 6 foods-10-01715-t006:** Results of DCC-GARCH models.

	ARM	BRA	CMR	GEO	JPN	KGZ	KOR	MEX	PER	ZAF
Model	AR-ADCC	ADCC	AR-ADCC	DCC	DCC	ADCC	AR-DCC	AR-ADCC	AR-DCC	ADCC
*λ* _1_	0	0	0.008	0	0.016	0.094	0.070	0	0	0
	(0.058)	(0.372)	(0.028)	(0.703)	(0.023)	(0.062)	(0.092)	(0.266)	(0.024)	(0.098)
*λ* _2_	0.856 ***	0.342	0.877 ***	0.703 **	0.911 ***	0.252	0.442	0	0	0.422 ***
	(0.33)	(0.809)	(0.052)	(0.299)	(0.055)	(0.271)	(0.66)	(1.888)	(0.268)	(0.195)
*δ*	0.02	0.26		0.13				0.33	0.18	0.27
	(0.067)	(0.585)		(0.273)				(0.349)	(0.263)	(0.329)
Log Likelihood	−782.55	−930.92	−859.18	−868.45	−825.49	−851.38	−832.78	−831.05	−688.17	−872.00
BIC	11.35	13.34	12.31	12.48	11.85	12.20	12.02	12.02	10.04	12.53

Note: *** *p* < 0.01, ** *p* < 0.05. The values in parentheses are standard errors.

**Table 7 foods-10-01715-t007:** Descriptive statistics on the estimated DCC.

Variables	n	Mean	Standard Deviation	Median	Minimum	Maximum	Range	Skewness	Kurtosis	Standard Error
ARM	144	0.072	0.012	0.068	0.06	0.128	0.068	1.794	3.987	0.001
BRA	144	0.148	0.078	0.117	0.058	0.567	0.509	2.855	9.946	0.006
CMR	144	0.306	0.027	0.301	0.24	0.409	0.169	1.384	3.067	0.002
GEO	144	0.16	0.043	0.151	0.08	0.314	0.234	0.903	0.919	0.004
JPN	144	0.063	0.037	0.061	−0.06	0.151	0.211	−0.738	1.513	0.003
KGZ	144	0.24	0.082	0.243	−0.042	0.709	0.751	1.172	8.623	0.007
KOR	144	0.193	0.054	0.195	−0.049	0.402	0.451	−0.29	4.247	0.004
MEX	144	0.093	0.093	0.064	0.028	0.788	0.759	4.25	22.887	0.008
PER	144	0.244	0.056	0.229	0.132	0.571	0.439	3.023	11.968	0.005
ZAF	144	0.28	0.083	0.255	0.127	0.782	0.655	2.617	10.509	0.007

**Table 8 foods-10-01715-t008:** Results of panel data analysis.

	(1)	(2)	(3)	(4)
	Fixed Effect	Fixed Effect	Random Effect	Random Effect
*Dependence*	0.1209 ***	0.1213 ***	0.1147 ***	0.1151 ***
	(0.0395)	(0.0422)	(0.0375)	(0.0404)
*GDP*	−0.1337	0.1119	−0.1223	−0.0015
	(0.3522)	(0.3668)	(0.2847)	(0.2958)
*CPI*	1.1309 ***	1.3106 ***	0.9090 ***	1.0061 ***
	(0.4096)	(0.4232)	(0.3493)	(0.3604)
∆*er*	−2.7150	3.3906	8.3880	13.1220
	(14.9728)	(15.8401)	(10.1997)	(10.7663)
∆*SC_barley_*		−0.0419		−0.0296
		(0.0390)		(0.0360)
∆*SC_maize_*		0.0063		0.0161
		(0.0489)		(0.0452)
∆*SC_rice_*		0.1299 *		0.0680
		(0.0740)		(0.0544)
Constant			6.3294 *	5.1448
			(3.4526)	(3.7101)
Year dummy	Yes	Yes	Yes	Yes
Observations	100	90	100	90
R^2^	0.1212	0.1883	0.1123	0.1583
Adjusted R^2^	−0.0116	0.0237	0.0750	0.0864
F Statistic	2.9664 **	2.4521 **	12.0233 **	15.4198 **

Note: * *p* < 0.1, ** *p* < 0.05, *** *p* < 0.01.

## Data Availability

The dataset used in this research is available in the Global Information and Early Warning System: https://fpma.apps.fao.org/giews/food-prices/tool/public/#/home and Food and Agriculture Organization STAT: http://www.fao.org/faostat/en/#data.

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
