# Peer review of "Food Import Dependency and National Food Security: A Price Transmission Analysis for the Wheat Sector"

_foods, 2021, doi:10.3390/foods10081715_

Round 1

Reviewer 1 Report

Let me first say that unfortunately, I don't have particular expertise on the methodology used because my work as a researcher, at least in the field of econometrics, is confined to the use of cross sections. Nonetheless, I read with some interest this manuscript on food security and the transmission of price volatility in the case of wheat. The general topic is of great interest also because dependence on foreign markets, price volatility of the most important commodities and food security in general are very much at risk during this pandemic period. However, I have many concerns especially about the effectiveness of discussions organised by the authors in the form of policy implications. Some methodological aspects are also not entirely clear to me.

Here are my main observations and concerns:

It is not clear to me what criteria the authors adopted to select the countries under investigation. For example, why were no MENA countries considered where food riots developed as a result of rising international wheat prices? From reading lines 150 - 158 it seems to me that the countries chosen are the only ones for which the necessary data were available. This motivation seems very weak. Furthermore, the authors claim to have collected prices from January 2005 to December 2019 for eight wheat-importing countries. Soon after a list of 11 countries on which the analysis is carried out is showed. Finally the authors talk about ten regions for which data is available.This is an important aspect of the paper and I invite the authors to carefully explain the reasons behind the choice of these countries and to clarify the actual group of countries

On line 75 the authors present the organisation of the paper. I recommend using the section titles actually used and then replacing the expression Methodology with Materials and Methods

The section on policy implication looks very weak. The opening sentence: "Our results indicate that reducing the foreign dependency on wheat is statistically significant" don't have any logical sense. What does it mean that reducing the foreign dependency on wheat is statistically significant?

Much of the policy implications are devoted to substituting cereal consumption as a means of mitigating the transmission of price volatility from the world market. But this consideration does not follow directly from the results. Moreover this consideration is partially contradicted by the statement made in the conclusions :”substitutive grain commodities such as  barley, maize and rice do not absorb jolts from overseas markets”. Furthermore, the final remark on the 4 pillars of food security is not related to the results. Why should reducing food dependency increase availability and stability? What elements of the analysis allow us to say that utilisation would not be compromised? Finally, the reference to the problem of access seems to me quite obvious and again not related to the results.

Author Response

Thank you very much for all of your comments. Please see the attached file for detailed responses. 

Reviewer 2 Report

This paper analyzes whether wheat import dependency could make a country vulnerable to overseas shocks in 10 countries for the 2005-2019 period.

Results show that retail price volatility in each region was positively correlated with international price volatility, demonstrating that external dependency could significantly protect the domestic market from the global one.

It is an interesting issue for Food journal, but authors should improve paper as follows:

  1. Authors have to provide the keywords in their paper.
  2. In the Introduction section, authors have to add that food prices depend also by energy crops, especially by first generation ones (i.e. mais) that, by subtracting agricultural areas to food products affect significantly the agricultural markets. In economic literature there is a large debate on arable land change.
  3. Authors should create a Discussion section, moving into it the “policy implications” section and taking into consideration also the entrepreneurial implications (what are the impact of price volatility for farmer?). In this section, authors have to compare their results with other scientific papers extant in the economic literature.
  4. References section is too poor.

I don't feel qualified to judge about the English Language and Style.

So, I recommend with major revisions.

Author Response

Thank you very much for all of your comments. Please see the attached file for detailed responses.

Many thanks.

Round 2

Reviewer 1 Report

The authors correctly answered to all points raised and the paper in present form is sufficiently improved. I believe it is ready for publication.

Author Response

Thanks

Reviewer 2 Report

The authors have addressed fruitfully every my suggestion.

Author Response

Thanks